# Strategies to Increase Flu Vaccination Coverage among Healthcare Workers: A 4 Years Study in a Large Italian Teaching Hospital

**DOI:** 10.3390/vaccines8010085

**Published:** 2020-02-13

**Authors:** Andrea Barbara, Daniele Ignazio La Milia, Marcello Di Pumpo, Alessia Tognetto, Andrea Tamburrano, Doriana Vallone, Carlo Viora, Silvia Cavalieri, Andrea Cambieri, Umberto Moscato, Filippo Berloco, Gianfranco Damiani, Walter Ricciardi, Giovanni Capelli, Patrizia Laurenti

**Affiliations:** 1Section of Hygiene, Institute of Public Health, Università Cattolica del Sacro Cuore, 00168 Rome, Italy; andreabarbara89@hotmail.it (A.B.); dipumpomarcello@gmail.com (M.D.P.); alessia.tognetto@gmail.com (A.T.); dott.tamburrano@gmail.com (A.T.); dvmedi@gmail.com (D.V.); umberto.moscato@unicatt.it (U.M.); gianfranco.damiani@unicatt.it (G.D.); walter.ricciardi@unicatt.it (W.R.); patrizia.laurenti@unicatt.it (P.L.); 2Hospital Hygiene Unit, Fondazione Policlinico Universitario “Agostino Gemelli” IRCCS, 00168 Rome, Italy; 3Section of Occupational Medicine, Institute of Public Health, Università Cattolica del Sacro Cuore, 00168 Rome, Italy; carloviora@hotmail.it (C.V.); silviacavalieri23@gmail.com (S.C.); 4Fondazione Policlinico Universitario “Agostino Gemelli” IRCCS, 00168 Rome, Italy; andrea.cambieri@policlinicogemelli.it (A.C.); filippo.berloco@policlinicogemelli.it (F.B.); 5Department of Human Sciences, Society and Health, Università degli Studi di Cassino e del Lazio Meridionale, 03043 Cassino, Italy; g.capelli@unicas.it

**Keywords:** flu vaccination, healthcare workers, teaching hospital, on-site vaccination, preventive medicine, vaccination coverage

## Abstract

Flu vaccination is recommended among healthcare workers (HCWs). The low vaccination coverage registered in our hospital among HCWs called for new engaging approaches to improve flu vaccination coverage. The aim of this study was to evaluate the efficacy of different strategies implemented during the last four years (2015–2019). A quasi-experimental study was conducted, involving almost 4000 HCWs each year. Starting from the 2015–2016 campaign, new evidence-based strategies were progressively implemented. At the end of each campaign, an evaluation of the vaccination coverage rate reached was performed. Moreover, during the last three campaigns, differences in coverage among job category, wards involved or not in on-site vaccination (OSV) intervention, age classes and gender were analyzed. An increasing flu vaccination coverage rate was registered, from 6% in 2015–2016 to almost 22% at the end of 2018–2019. The overall number of vaccinated HCWs increased, especially at younger ages. OSV strategy always leads to better results, and physicians always show a higher vaccination coverage than nurses and other HCWs. The implemented strategies were effective in achieving higher flu vaccination coverage among HCWs in our hospital and therefore can be considered valuable examples of good prevention practices in hospital settings.

## 1. Introduction

Influenza is an infectious disease with a high impact in terms of incidence, morbidity and mortality [1,2,3]. It is responsible for an increase in hospitalizations and direct and indirect costs all over the world, thus representing a serious public health issue [1,4,5,6]. Seasonal flu vaccination is one of the most important strategies for preventing influenza and reducing its healthcare, social and economic impact [5,7,8,9]. According to the World Health Organization (WHO) recommendations, at least 75% (optimal level: 95%) of at-risk individuals, including Healthcare Workers (HCWs), should be vaccinated in order to optimize the prevention and control of influenza and to reduce its impact [10]. In Italy, this recommendation has been included by the Ministry of Health in the National Vaccination Plan 2017–2019 (Piano Nazionale Prevenzione Vaccinale, PNPV) [11] and in a national law concerning health and safety at work [12]. These norms recommend, among HCWs exposed to biological agents, general and special protective evidence-based measures to prevent the transmission of influenza: washing and cleaning hands frequently, maintaining good respiratory hygiene, using facemasks and undergoing seasonal flu vaccination [11,12]. During the influenza season, in fact, HCWs can be affected by clinical and sub-clinical (oligosymptomatic or asymptomatic) flu and may therefore transmit influenza virus to patients, increasing their length of hospital stay, the costs for testing and treatment and the risk of complications, which can also lead to severe consequences [13,14,15,16]. Vaccination among HCWs involved in the everyday care of patients limits the spread of the virus, ensuring healthcare assistance continuity, thanks to both the individual and herd immunity [13,14,17,18]. Moreover, it reduces absenteeism [8,19,20].

Despite this, the vaccination coverage rate among HCWs is usually lower than what is recommended [10,21,22] for several reasons [23,24,25,26], even if rates achieved among different countries are frequently divergent. For instance, during the 2017–2018 season, in the USA, where vaccination is mandatory for HCWs, the coverage rate reached more than 90% among HCWs in hospital settings [27]. In Europe, where vaccination is not mandatory, the median HCWs vaccine coverage rate in 2016–2017 was 30.2% [22]. In Italy, the mean national influenza vaccination coverage among HCWs was 15.6% during the 2016–2017 influenza season [22], with huge differences among regions, hospitals and HCWs professionals [19,28,29,30,31]. According to two systematic reviews, in Italy, it is possible to estimate the influenza vaccination coverage close to 13% among nurses and 23% among physicians [32,33].

In our Fondazione Policlinico Universitario “A. Gemelli” IRCCS (FPG) hospital, a very low flu vaccination coverage rate among HCWs was registered during the 2014–2015 season [34]. Based on this result, increasing the vaccination coverage rate for influenza vaccination among HCWs represented (and still represents) a hard but essential challenge calling for new engaging, evidence-based approaches. For this reason, before the 2015–2016 flu campaign, a new internal program started with the scope of increasing flu vaccination coverage through the implementation of several initiatives.

The aim of this study was to evaluate the overall efficacy of the different strategies implemented in the FPG teaching hospital during the 2015–2016, 2016–2017, 2017–2018 and 2018–2019 influenza seasons to increase flu vaccination coverage among HCWs.

## 2. Materials and Methods

### 2.1. Setting, Study Design and Population

A quasi-experimental study was conducted in the Fondazione Policlinico Universitario Agostino Gemelli IRCCS (FPG), a large Italian Teaching Hospital located in Rome—Regione Lazio composed by two main buildings (the Policlinico Gemelli itself and the Presidio Columbus—CIC, a minor building nearby). It represents a regional and national excellence with 1500 beds and over 5500 employees. About 4000 of them are HCWs with daily contact with patients.

### 2.2. Implemented Strategies

Before the beginning of the 2015–2016 flu season, a multi-professional group focused on influenza vaccination was created to develop a “long-term, step-by-step” project aimed at increasing flu vaccination coverage rates among HCWs. Firstly, a literature search on vaccine hesitancy among HCWs and an internal audit with hospital key opinion leaders were performed. Concerns about safety, necessity and effectiveness of influenza vaccination, the lack of awareness of being potentially responsible for the transmission to patients, the lack of time and absent or inadequate vaccination policies and services were identified as main reasons to delay or refuse flu vaccination, both in the literature and in our setting [24,25,35,36,37,38]. Based on these evidences, starting from the 2015–2016 influenza season, different evidence-based strategies [24,39,40,41,42,43,44,45] were progressively implemented in the following seasons (Table 1).

### 2.3. Forum Theater (FT)

FT is a peculiar methodology used, especially in social and political settings, with the purpose of promoting knowledge, attitude and practice changes [46]. It is characterized by a participative and pro-active approach to discuss a problem, in which the audience (the “spect-actors”) is involved in the plot and encouraged to suggest different resolution for the play and to directly come on stage and act [46]. Although already implemented in communities and hospital setting, to our knowledge, our experience represented the first application in the field of flu vaccination among HCWs [42].

### 2.4. The On-Site Vaccination (OSV) Intervention

This new bottom-up approach has been carried out since the 2016–2017 campaign. This strategy emerged as request from HCWs who participated at the FT [42]. Thought, this project has been already used elsewhere, as reported in the scientific literature [39,47,48,49,50,51]. The OSV consisted of personal visit by 2 trained medical residents in Hygiene and Preventive Medicine (Public Health) and/or Occupational Medicine to wards to perform influenza immunization counseling and to vaccinate free of charge HCWs who wanted to be vaccinated. The date of each OSV intervention was previously settled according to the director of the operative unit and the nurse coordinator of the ward involved. Every OSV session was preceded by an email remind directed to the nursing coordinator. During the 2018–2019 season, nursing students were involved in performing the OSV visits in addition to medical residents.

#### Selection Criteria for the OSV Intervention Implemented during the Different Influenza Campaigns

Before the 2016–2017 influenza season campaign, the operative units and wards of the FPG were grouped in 36 macro-areas, according to clinical and functional characteristics (e.g., all cardiovascular wards were grouped in one macro-area; hematology, radiotherapy and medical oncology wards were grouped in the onco-hematology macro-area). For the 2016–2017 campaign, considering the healthcare workforce available to perform the OSV intervention, a random selection of 12 out of 36 macro-areas (33%) was done, using a random number generator without repetition in Excel. In the 2017–2018 season, the OSV intervention was extended to the wards nearby the ones involved during the previous campaign, utilizing therefore a “geographical” criterion (proximity) rather than the “functional” one used in the first year. During the last flu vaccination campaign (2018–2019 season), all the hospital wards were involved in the OSV intervention; therefore, no wards were available to be used as control (no-OSV) for the analysis.

### 2.5. Data Collection, Management and Ethical Committee

Data such as age, gender, professional category, and being vaccinated during the campaign and in the previous season (from 2017–2018 campaign) were collected and stored in an electronic database, accessible by password only to the data manager. In particular, to register vaccination status as “yes”, only the vaccinations administered by FPG staff inside the hospital were considered. HCWs were grouped per job categories in medical doctors (physicians), nurses and other HCWs (OHCWs; e.g., obstetricians, physiotherapists, etc.). Variable “age”, collected as continuous data, was subsequently grouped in 5 classes: min/34; 35/44; 45/54; 55/64; and 65/max. These class cutoffs were chosen because, in Italy, flu vaccination is recommended and offered for free to all the people over 65 years [11]. Moreover, during the 2018–2019 season, the role played by the leaders (both nurse coordinators and head physicians of each ward involved) in increasing the vaccination coverage of the HCWs directly employed was investigated, analyzing the vaccination coverage rate of the HCWs with a vaccinated leader compared to that one of the HCWs with a non-vaccinated leader.

The study was conducted in accordance with the Declaration of Helsinki, and the protocol was approved by the Ethics Committee the FPG Teaching Hospital—Università Cattolica del Sacro Cuore, Rome (Reference n. 47616/17 ID 1786 and n. 41409/18 ID 2263).

### 2.6. Statistical Analysis

At the end of each influenza season (2015–2016, 2016–2017, 2017–2018 and 2018–2019), a cross-sectional study was carried out to evaluate the vaccination coverage rate reached and the overall efficacy of the implemented strategies. The 2015–2016 vaccination coverage was used as the “starting point” to evaluate the effectiveness of the strategies implemented during the following years, even though it was partially influenced by some of them, in particular by the FT (which, nevertheless, was not implemented again in the following campaigns). Data collected were summarized in terms of absolute and relative frequencies or mean and standard deviation (SD) for qualitative and quantitative variables, respectively. Statistically significant differences among groups were tested through χ^2^ test or *t*-test, where applicable. The percentage variations among seasons (2016–2017 and 2017–2018; 2017–2018 and 2018–2019) were calculated to better analyze changes before and after each campaign. A multivariate logistic regression analysis was performed to assess differences in flu vaccination coverage between groups for each of the last three campaigns (2016–2017, 2017–2018 and 2018–2019). The variables included in the multivariate analysis were age class, gender, job category and OSV. In the logistic analysis, we decided to use the discrete age classes instead of the continuous variable, because the continuous variable showed departure from linearity in the models. Statistical significance was set at 0.05. Analyses were performed by using STATA 15.1 Software (StataCorp, Texas).

## 3. Results

Starting from the 2015–2016 campaign, an increasing flu vaccination coverage rate among HCWs of the FPG teaching hospital was registered. Characteristics of the population investigated during the 3 following seasons (2016–2017, 2017–2018 and 2018–2019) are shown in Table 2; the results of the multivariate analysis (one for each flu vaccination campaign) are presented in Table 3.

### 3.1. The 2015–2016 Flu Vaccination Campaign

At the end of the 2015–2016 season, the vaccination coverage rate against influenza among HCWs was almost 6%. Moreover, more than 74% of the participants were satisfied about the FT experience and 70% of those who were interviewed considered this methodology a useful approach for other health issues [42].

### 3.2. The 2016–2017 Flu Vaccination Campaign

During the 2016–2017 campaign, the OSV intervention was performed in 12 of the 36 (33%) macro-areas, for a total of 1120 out of 3654 (30.7%) HCWs potentially involved in OSV (Appendix A). The overall vaccination coverage rate among HCWs was 9.30%, significantly higher than that one reached in 2015–2016 (*p* < 0.0001). In 9 out of 12 (75%) macro-areas in which OSV was performed, the vaccination coverage among HCWs was higher than the overall coverage reached in all the hospital (Appendix A). The overall flu vaccination coverage in HCWs working in wards involved in OSV intervention was 14.0% vs. 7.2% among HCWs not interested by OSV (*p* < 0.001). The mean age ± SD among vaccinated vs. non-vaccinated HCWs was 49.4 years ± 10.3 vs. 44.3 ± 10.3 (*p* < 0.001). The highest vaccination coverage rate was in the age class of 65 years old or more, the lowest in the youngest age class.

According to the multivariate analysis, age classes 45–54, 55–64 and more than 64 years old, male sex and working in wards reached by OSV were significantly associated with a higher influenza vaccination adherence. Moreover, being nurse or OHCWs was statistically related with lower influenza vaccination coverage in comparison with physicians.

### 3.3. The 2017–2018 Flu Vaccination Campaign

The flu vaccination campaign approached 3664 HCWs, including 1960 nurses, 1015 physicians and 689 OHCWs. The overall vaccination coverage during this campaign was 14.0% (*n* = 514). This result was significantly higher than the one achieved in 2016–2017 (*p* < 0.0001). The mean age ± SD among HCWs vaccinated was higher than non-vaccinated (*p* < 0.001). As in the previous season, the highest vaccination coverage rate was among HCWs of 65 years old or more, the lowest in the youngest age class even though this last was the age class with the highest increase from the previous season. The vaccine uptake was higher among men (20.1% vs. 10.8% in females; *p* < 0.001). Physicians were the most inclined professionals to undergo vaccination (25.2% of their category), while the OHCWs were the professionals with the highest increase (+105.9%), followed by nurses (+71.1%). The OSV sessions were extended to the wards nearby the ones involved during the previous year, potentially involving 1858 HCWs, more than half (50.7%) of the total HCWs (+65% compared to the previous season). Among the HCWs involved by the OSV intervention, vaccination coverage rate was 19.1% vs. 8.9% among HCWs not potentially involved (*p* < 0.001). Moreover, it seems that the vaccination status of the previous season had an impact on undergoing flu shot (Table 2).

According to the multivariate analysis, belonging to older-age classes (45–54 years old, 55–64 years old and more than 64 years old), being male and working in wards reached by the OSV were statistically significant in increasing influenza vaccination adherence. Instead, being nurses or OHCWs was statistically significant in lowering influenza vaccination coverage in comparison with physicians.

### 3.4. The 2018–2019 Flu Vaccination Campaign

During the last influenza season considered in the study (2018–2019), the overall vaccination coverage rate among HCWs was 22.0%, significantly higher than that one previously reached (*p* < 0.001). In the univariate analysis, vaccine uptake was significantly related to age class, with a decreasing trend (in the older age class, equal to −29.3%). Moreover, a statistically significant difference in vaccination coverage among professional categories was confirmed, with the highest increasing registered among nurses (+80.5%). Furthermore, statistical differences in flu vaccination coverage were reported when a comparison of the vaccination status during the two previous seasons (2016–2017 and 2017–2018) was done. Concerning the role played by the leaders, it seems that the HCWs with a vaccinated leader showed a higher vaccination coverage rate than the HCWs with a non-vaccinated leader (*p* < 0.001).

The results of a multivariate analysis showed that the 55–64 years old age class was statistically significant in influencing vaccination coverage during the 2018–2019 season in comparison with the youngest age class (aOR 1.58, 95% CI 1.25–1.99), such as being nurse (aOR 0.42, 95% CI 0.35–0.50) or OHCWs (aOR 0.38, 95% CI 0.31–0.47) in comparison with physician.

### 3.5. Overall Strategies Effectiveness

To evaluate the progression in the overall effectiveness of the following vaccination campaigns, we can compare the absolute number of vaccinated and non-vaccinated HCWs by year of age (Figure 1) or the vaccination coverage by 10-year age classes (Figure 2).

What clearly emerges from Figure 1 is that, not only the overall number of vaccinated HCWs increased (as already stated in the first sections of the results), but the increase was higher at younger ages.

This result is confirmed in Figure 2, in which the vaccination coverage by gender, job category and age class presented in Table 3 are shown. On-site vaccination strategy always leads to better results (orange lines), physicians always show higher vaccination coverage rates than nurses and OHCWs, but the progression among the three campaigns is particularly accentuated among professionals aged <45, which in on-site wards add an extra 5%–10% of vaccination coverage at each campaign.

Finally, Appendix A show the probability of being vaccinated, calculated using the same logistic model presented in Table 3 (S1) and using a different one (S2), which introduces a multiplicative interaction among HCW group, gender and age class (Appendix A).

## 4. Discussion

Starting from the 2015–2016 season, an increasing overall flu vaccination coverage rate trend among HCWs was registered in the FPG teaching hospital, from 6% coverage in 2015–2016 to almost 22% at the end of the 2018–2019 season. One of the reasons of this increase was the implementation of the multicomponent strategy including the education of HCWs about vaccination, the active promotion of the vaccination campaign and the easy access to influenza vaccination free of charge, obtained mainly thanks to OSV intervention. Nevertheless, the overall vaccination coverage rate was lower than those reached in other international settings [22,27,29,51,52], thus remaining unsatisfactory and still far away by the minimum goal of 75% defined by the WHO and by the Italian PNPV (PNPV 2017–2019) [10,11]. However, the vaccination coverage registered in our teaching hospital represents an encouraging starting point and it is analogous, or even higher, than flu vaccination coverage rates registered in other national and regional settings [29,32,33,39,50,53,54,55]. Our results confirmed that physicians are the professionals most willing to get vaccinated, as reported in the literature [20,33,50,54], even though the overall increase was the lowest during the considered period if compared to nurses and OHCWs. The job category that registered the highest increase in coverage rate (compared to the previous years) were the OHCWs (+76.3%) and the nurses (+75.8%), proving that the implemented strategies were effective in reaching groups originally less inclined to vaccination, maybe due to the wrong perception of being at low risk of illness or transmitting the infection. A similar phenomenon was registered considering gender, as males had the highest vaccination coverage, but with a lower increase than women during the flu vaccination campaigns. Among the five-age class considered, the highest vaccination coverage was observed in the oldest two age classes in each of the last three campaigns, while the lowest in the youngest one. Nevertheless, the vaccination coverage has grown more in younger-age classes. Moreover, the performed multiple interactions analysis show probability estimated “high steps” for the age class of the over 65 (over 55 in Males other HCW professionals), rather independent from the on-site vaccination proposal, almost certainly due to the over 65 awareness of their status of higher-risk population. However, the performance of the OSV still works at any age and shows the possibility to reach a coverage around 40%–50% in many HCWs/gender groups in the age range 55–65, still confirming a rise in the coverage among physicians and nurses, even at younger ages, compared to the not-on-site wards in the first 2016–2017 campaign (Appendix A).

Vaccine hesitancy is a complex issue, especially amongst HCWs, and it requires ad hoc studies to understand the phenomenon in different contexts, in order to find ad hoc effective strategies to struggle it [24,26,40,41,56]. First of all, even and especially among healthcare personnel, it is necessary a have a cultural change about knowledge, attitudes and practices on vaccination through personalized training, such as academic detailing (educational outreach visit) and problem-based learning methodologies [43,44] or through the role of the occupational doctor who carries out annual health surveillance of workers [12]. As a matter of fact, this figure could play a key role by counseling HCWs with adequate information on the advantages and disadvantages of vaccination and non-vaccination, as defined by the Italian law [12]. Second, it could be useful to demonstrate and sustain the economic value of influenza vaccination among HCWs, by performing an economic analysis related to sickness absenteeism that occurs during seasonal flu periods, as already done by ourselves [45] and elsewhere [8,19,55,57]. Furthermore, mandatory vaccination among HCWs may be evaluated, as proposed by other authors [26,41,56], and already applied in other countries (e.g., USA) and to other vaccines, such as in Italy on the pediatric population [58].

### Limitations

This study has several limitations. First of all, the exposure to the OSV intervention of HCWs pertaining to a specific macro-area is only “potential”: we assume that all of HCWs working in the ward involved in the OSV were present during the visit made by the trained medical residents that performed the OSV intervention, but there is no certain way to assess the presence of single HCWs at the time of every session. To limit this uncertainty, over the different flu vaccination campaigns, in many wards, more than one OSV intervention was carried out each year. Second, only the vaccinations administered inside the hospital were registered. Vaccination shots administered outside, by general practitioners or Local Health Authorities in primary care settings, were not collected/investigated. For this reason, the vaccination coverage rate registered among the HCWs in the FPG teaching hospital could be underestimated, overestimating, instead, the evaluation of the strategies implemented. This phenomenon, however, has less of an impact on the overall evaluation of the vaccination coverage rate than at the “starting point” of the program (2015–2016 season), thanks to the fundamental communication work done by the Health Management that every year spreads the news about the implemented strategies all over the hospital facilities, reducing the need to go outside the FPG to undergo flu vaccination.

## 5. Conclusions

Flu vaccination among HCWs is a fundamental measure to protect HCWs themselves, their patients and the hospital community. When low vaccination coverage is registered, new engaging approaches are necessary in order to sustain, encourage and promote the role of flu vaccination among HCWs. In our context, the OSV intervention and all the other strategies implemented resulted in being effective in achieving higher flu vaccination coverage and therefore can be considered a valuable example of good prevention practices in hospital settings. Moreover, though important, a single strategy has not proven to be sufficient to achieve the desired goal in vaccination coverage rates. For this reason, multiple approaches, including educational outreach visit, sickness absenteeism evaluation, cost-effective analysis, and mandatory vaccination and incentives or rewards to vaccinated HCWS, should be taken into consideration and assessed for implementation in different settings. Encouraging results in the younger age groups confirms the value of educational interventions.

## Figures and Tables

**Figure 1 vaccines-08-00085-f001:**
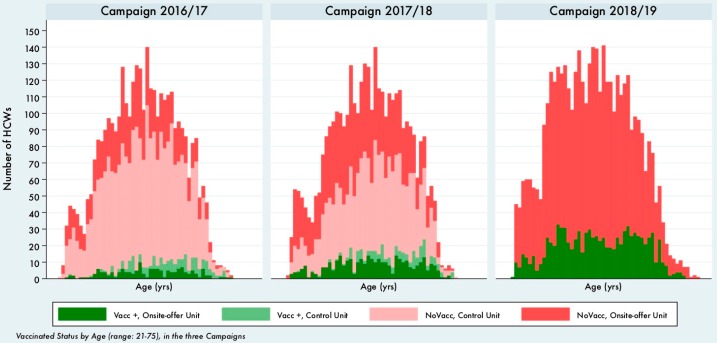
Vaccination status by age, campaign 2016–2017, 2017–2018 and 2018–2019.

**Figure 2 vaccines-08-00085-f002:**
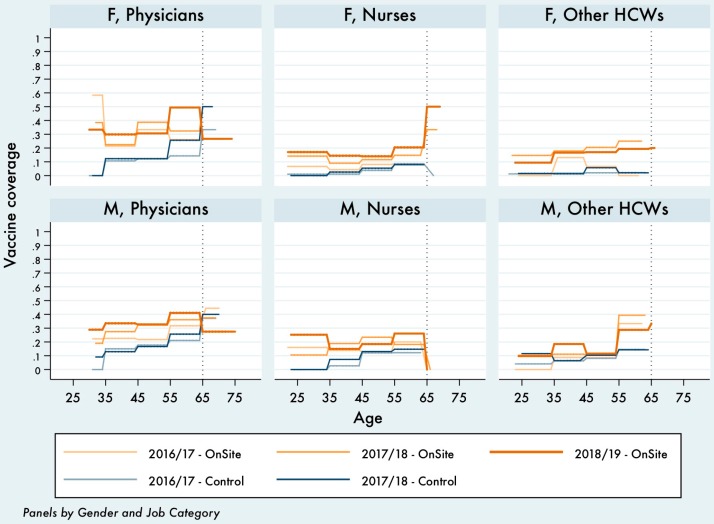
Vaccination coverage by age class, gender, OSV and job category during the three campaigns (model without interaction, available in Table 3).

**Table 1 vaccines-08-00085-t001:** Strategies implemented since 2015–2016 flu vaccination campaign.

Strategy	Description	Flu Vaccination Campaign			
		2015–2016	2016–2017	2017–2018	2018–2019
Forum Theatre (FT)	*Explained in the text*	x	-	-	-
Opening ceremony	Opening ceremony of the influenza season campaign targeted at all HCWs: the results of the previous campaign were shown and people in attendance were invited to adhere to the campaign.	x	>	>	O award ceremony of the ward with the highest flu vaccination coverage reached in 2017–2018
Opening hours extension	Opening hours extension of the ambulatory located in the teaching hospital that provided the vaccination service free of charge. Baseline: 1 h twice a week.	x	o 1 h and a half (from 1 p.m. to 2.30 p.m.) 3 days/week	o 2 h (from 12.30 a.m. to 2.30 p.m.) 5 days/week	>
Second ambulatory	Opening of a new ambulatory for vaccination, located in another strategic place to reach different population *(especially student, not involved in this study)*. Opening hours: 2 h 5 days per week.	x	>	>	>
E-mail invitation and reminders	Invitation and reminders via e-mail to undergo vaccination from the General Director and Chief Executive Officer, explaining the importance of the immunization among HCWs and the whole community plus specific information about where and when the internal vaccination service was provided.	x	o the OSV was included in information details	>	>
Promotional and informative material	Promotional and informative material including pins, flyers and advertising posters were placed in different strategic locations (e.g., main and secondary entrances, departments corridors, walls near to elevators).	-	x	o graphic changes	o further graphic changes
The on-site vaccination (OSV) intervention	*Explained in the text*	-	x	o ward increase	o all hospital wards involved
Academic Detailing	Also called educational outreach visit, university-based educational detailing or educational visiting is a peer-to-peer educational sessions [43] aimed at nurse coordinators in order to get them involved in promoting vaccination with their colleagues [44].	-	-	x	o frontal lessons removed; peer-to-peer interventions incremented
Non-economic incentive (one-hour leave)	A non-economic incentive, consisting in a one hour leave to all HCWs who got vaccinated against influenza. This was supported by an economic evaluation performed in our setting during the previous campaign that demonstrate the potential cost-effectiveness of this measure [45].	-	-	-	x

Legend: x. newly implemented; o. changed during the campaign considered in comparison with the previous one, followed by the explanation of the improvement done; >. maintained in the campaign as in the previous one; -. not implemented.

**Table 2 vaccines-08-00085-t002:** Characteristics of the HCWs and univariate analysis for the 2016–2017, 2017–2018 and 2018–2019 flu vaccination campaigns and percentage of increasing from 2016–2017 to 2017–2018 and from 2017–2018 to 2018–2019.

Variables	2016–2017				2017–2018				2018–2019				Δ% from 2016–2017 to 2017–2018	Δ% from 2017–2018 to 2018–2019
	Vaccinated	Total HCWs	%	*p*	Vaccinated	Total HCWs	%	*p*	Vaccinated	Total HCWs	%	*p*		
**Age class *§**														
≤34	31	653	4.75	<0.001	63	652	9.66	<0.001	163	868	18.78	<0.001	103.37	94.41
35–44	81	1085	7.47		125	1071	11.67		262	1216	21.55		56.22	84.66
45–54	98	1104	8.88		156	1142	13.66		230	1224	18.79		53.83	37.55
55–64	105	673	15.60		150	748	20.05		245	821	29.84		28.53	48.83
≥65	15	41	36.59		20	51	39.22		33	119	27.73		7.19	−29.30
**Gender ***														
F	140	2304	6.08	<0.001	257	2385	10.78	<0.001	533	2767	19.26	<0.001	77.30	78.66
M	190	1252	15.18		257	1279	20.09		400	1481	27.01		32.35	34.44
**Job Category**														
Physicians	197	1015	19.41	<0.001	256	1015	25.22	<0.001	448	1323	33.86	<0.001	29.93	34.26
Nurses	107	1959	5.46		183	1960	9.34		338	2005	16.86		71.06	80.51
OHCWs	36	680	5.29		75	689	10.89		147	920	15.98		105.86	46.74
**OSV**														
No	183	2534	7.22	<0.001	160	1806	8.86	<0.001					22.71	
Yes	157	1120	14.02		354	1858	19.05						35.88	
**Vaccinated during 2016–2017**														
No					279	3210	8.69	<0.001	505	3134	16.11	<0.001		85.39
Yes					218	326	66.87		244	313	77.96			16.58
Not present °					17	128	13.28		184	801	22.97			72.97
**Vaccinated during 2017–2018**														
No									412	3028	13.61	<0.001		
Yes									358	486	73.66			
Not present °									163	734	22.21			
**Leader Vaccinated**														
No									364	2256	16.13	<0.001		
Yes									569	1992	28.56			
**Total**	340	3654	9.30		514	3664	14.03		933	4248	21.96		50.86	56.52

Legend: HCWs: healthcare workers; OSV: on-site vaccination; OHCWs: other HCWs (e.g., obstetricians, physiotherapists, etc.). * Ten data were missing during the 2016–2017 season. ° newly hired, return from parental leave or time off work. ^§^ Considering age as a continue variable the mean age ± SD among vaccinated vs. non-vaccinated during the different seasons was 49.4 ± 10.3 vs. 44.3 ± 10.3 (*p* < 0.001) in 2016–2017; 48.2 ± 10.5 vs. 44.7 ± 10.9 (*p* < 0.001) in 2017–2018; 46.3 ± 11.6 vs. 44.4 ± 10.9 (*p* < 0.001) in 2018–2019.

**Table 3 vaccines-08-00085-t003:** Logistic regression (one for each flu vaccination campaign). Vaccination (yes/no) during the relative influenza season is the dependent variable.

Variables	2016–2017			2017–2018			2018–2019		
	aOR	95% CI	*p*	aOR	95% CI	*p*	aOR	95% CI	*p*
**Age class**									
≤34	1			1			1		
35–44	1.19	0.77–1.85	0.44	1.00	0.71–1.39	0.99	1.00	0.80–1.25	0.99
45–54	1.69	1.10–2.58	0.02	1.41	1.02–1.94	0.04	0.96	0.77–1.21	0.74
55–64	2.41	1.55–3.73	<0.001	1.93	1.38–2.70	<0.001	1.58	1.25–1.99	<0.001
≥65	4.20	1.94–9.10	<0.001	3.07	1.59–5.94	0.001	0.86	0.55–1.36	0.53
**Gender**									
F	1			1			1		
M	1.61	1.24–2.09	<0.001	1.31	1.05–1.62	0.02	1.12	0.95–1.33	0.163
**Job Category**									
Physicians	1			1			1		
Nurses	0.33	0.24–0.44	<0.001	0.37	0.30–0.48	<0.001	0.42	0.35–0.50	<0.001
OHCWs	0.31	0.21–0.46	<0.001	0.45	0.34–0.60	<0.001	0.38	0.31–0.47	<0.001
**OSV**									
No	1			1			*all hospital wards were involved in OSV*		
Yes	2.30	1.80–2.92	<0.001	2.64	2.15–3.25	<0.001			

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
