# Peer review of "Strategies to Increase Flu Vaccination Coverage among Healthcare Workers: A 4 Years Study in a Large Italian Teaching Hospital"

_vaccines, 2020, doi:10.3390/vaccines8010085_

Round 1
Reviewer 1 Report
The manuscript by Barbara et al sought to determine intervention strategies to increase flu vaccine compliance within a hospital. The data is interesting despite many country health systems have adopted mandatory vaccination for healthcare workers. I would suggest the authors edit their manuscript a little more closely as some of the sentences are not logically structured or are run ons.
Examples:
Line 306...sentence should be changed.
Line 264...rising up
Line 202...incline should be inclined
These are just a few examples of a need for better proof-reading.
Reviewer 2 Report
In this paper, Barbara et al. reported vaccination coverage among healthcare workers in a hospital in Italy and its risk factors including interventions they implemented. Although the findings were not novel for the topic that had been intensively investigated in many previous studies, their large-scale analysis at a single hospital may make their work worth to publishing. Yet, I have several concerns before the publication. Major points (Figures 2 and3, and lines 230-260) Although authors calculated probability using logistic regression model, the title of y-axis said “Vaccination coverage”. Prediction probability (by logistic regression model) applies to individuals with determined variable to predict outcome (= vaccination status). That cannot be used to predict vaccination proportion (= coverage) in population of the individuals. I would rather want to see actual vaccination coverage observed in the study in the Figures 2 and 3. Prediction probabilities were not informative for those figures. (Table 3 is enough to mention the logistic regression analysis.) (Line 68) Reference for their previous observation (low vaccination coverage in 2014-2015 season) should be cited. If there is no source for the result, authors might want to include the point in the Results. (Results and Discussion) Although authors mentioned other interventions than OSV listed in Table 1 in the Limitations and Conclusion, I would like the authors to analyze and/or discuss the impact of each of non-OSV interventions concretely in the Results and/or Discussion. Minor points (Throughout the manuscript) There were grammatical errors in the manuscript. Their English should be reviewed and edited by native English speaker. (Line 38) What is “National Health Services all over the world”? There is no standardized “national health services” among countries. Even some countries do not have “national” health services. Authors should change wording to describe economic burden of influenza at global scale. (Line 55) What is “collective protection”? Is it something like “herd immunity”, or “indirect protection”? (Lie 103) Although “FT” was explained in Table 1, authors should describe this in the main texts as well. (Figure 1-3) Green indicates vaccination and red indicates no-vaccination in Figure 1 whereas green indicates On-site-vaccination and red indicates control in Figures 2 and 3. This is really confusing. For the ease of readers, I would recommend authors use dark-green for ‘vaccination, OSV’, light-green for “no-vaccination, OSV”, dark-red for ‘vaccination, control’, and light-red for ‘no-vaccination, control’ in Figure 1 to make the figure concordant with Figures 2 and 3. (Figure 1) Indices for x-axis and titles for x-axis and y-axis should be inserted. Actually, x-axis (age) can be grouped as the authors did in Figures 2 and 3. (Line 287) Why “he”? There were no female doctors in the hospital?Author Response
Please see the attachment

Reviewer 3 Report
Thank you for the opportunity to review this interesting research paper. This paper considers the effectiveness of interventions implemented sequentially over several flu seasons at a teaching hospital in Italy. The results could provide interesting and useful information regarding how effective programs are at increase vaccination coverage among HCWs. The pseudo-experimental design provides evidence (albeit weak evidence) regarding the effectiveness of interventions. However, the study design could have been improved with a true comparison group (using a controlled-before-after design) for all stages and aspects of the implemented interventions. Nevertheless, for one flu season the onsite vaccination intervention was implemented in a random selection of wards and compared to wards where it was not yet implemented.
The paper is structured nicely and the tables include a good selection of information. The English needs to be checked. Some sentences are difficult understand (grammar/sentence structure) and there are minor spelling errors throughout the text.
I do have a few comments and questions as well as a few minor issues that need to be addressed:
As stated above, the first year that on-site vaccinations were provided to a random sample of wards could be considered controlled before-after study. The evidence provided by this research would be greatly improved if the data from this wave was analyzed as a cluster-randomized intervention study. The number of clusters (wards) and the size of the clusters should also be reported. Figures 1-3: The figures are attractive but difficult to understand. Not sure what they depict or if the information depicted is useful. Perhaps they should be moved to supplementary information? What reasoning do you give for considering the interactions shown in the supplement? Rate vs. coverage: The term rate is used to describe the frequency within a certain time-frame. While vaccination coverage in a flu season could possibly be considered #vaccinations/population*1 year (vaccination/people-years) you technically do not report the results as rates. So the term “vaccination rates” seems misleading and should be avoided. The fact that people who were vaccinated elsewhere (privately) were not assessed is a major limitation and listed as such in the discussion. However, how vaccination coverage was assessed should be made clearer in the methods, so the reader understands that vaccination coverage may be underestimated due to the lack of this information while reading the results. Could the overall percent (and number) vaccinated prior to the first intervention season considered be mentioned in the text? Do you have any indication how effect the first interventions were compared to the on-site vaccinations? Could you provide more details regarding the FT intervention? Perhaps an example of what it entails? Table 1: Other than FT, no papers are cited for the other interventions. Could references be included, so interested readers could find more details on the interventions- Line 182-183: The fact that 57.9% of vaccinated HCSs were physicians is not really informative or interesting. The vaccination coverage among job categories, as shown elsewhere in the paper is more informative, regarding how effective the interventions are among different groups of HCWs. Figures 1-3: Are the age-groups equivalent to birth/age cohorts, so that the vaccination coverage during the flu seasons are always being compared among the same group (within the same birth cohort)? Does the OSV intervention seem to have less of an effect once people reach a certain age or among a group of people, for example born before 1950?
Minor issues:
Abstract: The abbreviation for the hospital (FPG) is not yet explained and should not be used in the abstract. Table 3: Adding a note to the lower right table cell would be helpful to remind readers that all wards received the on-site vaccination intervention in 2018/2019. Table 3: p-values are redundant. Reporting 95% confidence intervals should suffice. Line 181-182: It is unclear which mean age values correspond with which flu seasons. Line 227: I do not think the term “adherence” applies here. Since the results of a logistic regression is being described, the prevalence odds of being vaccinated in the 2018-2019 season was statistically higher among 55-64 year olds. Figure 1: The x-axis needs labels.Author Response
Please see the attachment

Round 2
Reviewer 2 Report
Most of the authors’ modification/correction sufficed my comments. However, there is one critical point that must be addressed.
Lines 402-403, 448-453
Although the authors replaced the Figure 2 with new one with corrected y-axis (Vaccination coverage) as I had suggested, they still described in the texts as: “the probability of being vaccinated…calculated through logistic regression models (Figure 2 and 3)” AND "Figure2, in which the vaccination coverage... calculated with the logistic model..."
Again, as I said in the first round of review, results of logistic model have nothing to do with (actual, or observed) vaccination coverage. To calculate the coverage, any statistical model is required. From the texts, I cannot tell if the authors correctly showed the data of vaccination coverage in the figure. Also, they forgot to delete the texts, “Figure 3”, which does not exist anymore.
Statement for the results of probability by the logistic model (lines 454-457) seemed fine, though.
Reviewer 3 Report
Thank you for your response. I see that a lot of thought has been made to the changes in the manuscript.
However, I do think that more thought is needed regarding whether the analysis is truly appropriate. I think it would be improved if considered with multi-level modelling (and the "macro areas" at least from 2016-17 and possibly 2017-18 analyzed as clusters). In effect, you did conduct a cluster-randomized study in 2016-2017, but you are not considering or reporting the results as such: "For the 2016-17 campaign, OSV was performed in 12 of the 36 macro areas (33%), identified through a random selection."
The low mobility of HCWs between macro areas could bias the results (presumably towards null effect), but this must be impacting the current results as well. Right now the data are considered independent when in fact the macro areas may not be truly independent. Right now I do not see the statistical analysis as being completely justified.
How random selection of macro areas was conducted (e.g. coin toss, random number tables) should be also mentioned.
The English of new or highly edited sections needs to be editing (for example, lines 219-225; 479-488).
Round 3
Reviewer 3 Report
Thank you for the changes. The reporting is greatly improved by the table of clusters and information about randomization method. The statistical analysis could still be improved with an analysis method for clustered data. But I suppose this may not impact the results much.